# Estrogens Counteract Platinum-Chemosensitivity by Modifying the Subcellular Localization of MDM4

**DOI:** 10.3390/cancers11091349

**Published:** 2019-09-12

**Authors:** Rossella Lucà, Giorgia di Blasio, Daniela Gallo, Valentina Monteleone, Isabella Manni, Laura Fici, Marianna Buttarelli, Germana Ciolli, Marsha Pellegrino, Emanuela Teveroni, Silvia Maiullari, Alessandra Ciucci, Alessandro Apollo, Francesca Mancini, Maria Pia Gentileschi, Gian Franco Zannoni, Alfredo Pontecorvi, Giovanni Scambia, Fabiola Moretti

**Affiliations:** 1Institute Cell Biology and Neurobiology, National Research Council of Italy (CNR), 00015 Monterotondo, Italy; 2Institute of Pathology, Università Cattolica del Sacro Cuore, 00168 Rome, Italy; 3Institute of Obstetrics and Gynecology, Università Cattolica del Sacro Cuore, 00168 Rome, Italy; 4Department of Woman and Child Health and Public Health, Fondazione Policlinico Universitario A. Gemelli, IRCCS, 00168 Rome, Italy; 5Department of Biology, University of Rome Tor Vergata, 00133 Rome, Italy; 6SAFU unit, Department of Research, Diagnosis and Innovative Technologies, Traslational Research Area, IRCCS Regina Elena National Cancer Institute, 00144 Rome, Italy; 7Unit of Cellular Networks and Molecular Therapeutic Targets, IRCCS Regina Elena National Cancer Institute, 00144 Rome, Italy; 8Institute of Anatomical Pathology, Università Cattolica del Sacro Cuore, 00168 Rome, Italy; 9Department of Gastroenterologic, Endocrine-Metabolic, and Nephro-Urologic Sciences, Fondazione Policlinico Universitario A. Gemelli, IRCCS, 00168 Rome, Italy; 10IRCCS Regina Elena National Cancer Institute, 00144 Rome, Italy

**Keywords:** estrogen, chemosensitivity, MDM4, sexual dimorphism, intracellular trafficking

## Abstract

Estrogen activity towards cancer-related pathways can impact therapeutic intervention. Recent omics data suggest possible crosstalk between estrogens/gender and MDM4, a key regulator of p53. Since MDM4 can either promote cell transformation or enhance DNA damage-sensitivity, we analysed in vivo impact of estrogens on both MDM4 activities. In Mdm4 transgenic mouse, Mdm4 accelerates the formation of fibrosarcoma and increases tumor sensitivity to cisplatin as well, thus confirming in vivo Mdm4 dual mode of action. Noteworthy, Mdm4 enhances chemo- and radio-sensitivity in male but not in female animals, whereas its tumor-promoting activity is not affected by mouse gender. Combination therapy of transgenic females with cisplatin and fulvestrant, a selective estrogen receptor degrader, was able to recover tumor cisplatin-sensitivity, demonstrating the relevance of estrogens in the observed sexual dimorphism. Molecularly, estrogen receptor-α alters intracellular localization of MDM4 by increasing its nuclear fraction correlated to decreased cell death, in a p53-independent manner. Importantly, MDM4 nuclear localization and intra-tumor estrogen availability correlate with decreased platinum-sensitivity and apoptosis and predicts poor disease-free survival in high-grade serous ovarian carcinoma. These data demonstrate estrogen ability to modulate chemo-sensitivity of MDM4-expressing tumors and to impinge on intracellular trafficking. They support potential usefulness of combination therapy involving anti-estrogenic drugs.

## 1. Introduction

Precision medicine is about matching safe and effective therapy to individual patients and requires knowledge of individual characteristics to define clinically relevant subgroups [1]. Gender is a factor that profoundly affects cancer incidence and susceptibility to therapy with sex hormones—especially estrogens—highly contributing to this disparity. Indeed, estrogens have well-recognized oncogenic properties in hormone-sensitive tumors. Accordingly, the estrogen receptor alpha (ERα) is an important therapeutic target in breast and endometrial cancer [2,3]. Additionally, estrogens can affect the p53 pathway. Estrogens enhance mRNA expression of MDM2, the major p53-inhibitor, by acting on the MDM2 promoter indirectly through AP1/ETS-responsive elements [4] or the SNP309 that confers binding to SP1 [5]. As confirmation of the relevance of this cross-talk, the SNP309G is associated with increased risk for hormone-sensitive tumors—breast, ovarian and endometrial cancer—and also “non-canonical” hormone-sensitive tumors—soft tissue sarcoma—in a gender- and hormone-dependent way [6,7].

Recent data from the Collaborative Oncological Gene-environment Study (COGS) identified SNP34091 in the 3′ UTR of the human MDM4 gene, as a cancer risk locus in hormonally mediated cancers [8,9]. MDM4 is the most proximal MDM2 homolog and also a relevant p53-regulator endowed of oncogenic activity and as such altered in different human tumors [10]. SNP34091 is assumed to alter MDM4 levels by creating a putative target site for hsa-miR-191, a microRNA highly expressed in tumor tissues [11,12]. This SNP confers increased cancer risk in ER-negative breast cancer [8,13] and prostate cancer [9], and is correlated with poor overall (OS) and disease-free survival (DFS) in ER-negative ovarian cancer [11]. These data suggest that the estrogenic pathway can interfere with MDM4 levels and/or function. However, the molecular basis of this interference remains elusive. Of note, in both ER-negative breast and ovarian cancer, the presence of this SNP is not correlated to the status of p53, suggesting that ER also interferes with MDM4 p53-independent activities. 

In fact, MDM4 possesses p53-independent oncogenic function [10]. Higher incidence of multiple tumors and a decreased animal survival in Mdm4 transgenic p53-null mice have confirmed its p53-independent tumorigenic potential [14]. Of note, these Mdm4-induced alterations are evident in p53-null males but not in females, indicating sexual dimorphism of Mdm4 activity, at least in rodents [14]. In human cells, MDM4 promotes degradation of the Rb protein [15]. 

On the other hand, in vitro experiments have demonstrated the existence of MDM4 “anti-tumor” activities under stress conditions: following DNA damage, MDM4 enhances cell apoptosis, by positively regulating p53 activity [16,17,18,19]. Furthermore, MDM4 inhibits metabolism [20] and counteracts mTOR [21]. In agreement with these “anticancer” activities, high levels of MDM4 correlate with increased sensitivity to platinum-based therapies in ovarian cancer [19] and better prognosis in breast cancer, independently of p53 status [22]. Mouse models to inform on MDM4 ability to alter chemotherapy sensitivity as well as the possible interference of gender/estrogens are lacking.

In this work, we have addressed the role of gender and estrogens in MDM4 activities by using transgenic mouse models overexpressing Mdm4. We show that in vivo Mdm4 is endowed both of tumorigenic potential by accelerating DNA-damage tumorigenesis, and anti-tumor activities by enhancing the sensitivity of fibrosarcoma to cisplatin and of thymocyte to radio-treatment. Of note, estrogens mediate sexual dimorphism in the sensitivity conferred by Mdm4 to DNA damage but not in its tumor-promoting function. 

## 2. Results

### 2.1. Enhanced Levels of MDM4 Accelerate DNA-Damage Induced Tumor Development in a Gender-Independent Manner

To study in vivo gender-dependent tumorigenic activity of MDM4, we used two transgenic mouse lines overexpressing different levels of Mdm4 (hereafter, Mdm4^Tg10^, and Mdm4^Tg15^ expressing low and high Mdm4 levels respectively) [23]. The transgene comprises only the cDNA of Mdm4, thus excluding the possible existence and interference in the murine gene of the SNP34091 described in the 3′UTR of the human gene. Considered the pro-apoptotic activity of MDM4 under DNA damage conditions [16,19], we induced primary tumors in transgenic mice by using the DNA-damaging agent, 3-methylcholantrene (3MCA), a polycyclic aromatic hydrocarbon with high carcinogenic properties. Since murine muscle expresses the estrogen receptor alpha (ERα) and high levels of the transgene Mdm4 (Appendix A) and is sensitive to 3MCA tumorigenic activity [24], we induced fibrosarcoma in the dorsal part of hindlimb muscle. The presence of the transgene significantly decreased the latency of tumor formation in Mdm4^Tg15^ mice compared to littermate control animals (WT) (Figure 1A,B), indicating that Mdm4 oncogenic properties prevail over its pro-apoptotic activities in this model. Distinct analysis of female and male groups did not evidence differences between two groups, indicating that the oncogenic properties of Mdm4 are not affected by the sex of the animals at least in these conditions. Accordingly, Mdm4 levels are similar in transgenic mice independently of the gender (Figure 1C). In Mdm4^Tg10^ mice, the lower levels of Mdm4 were unable to accelerate tumor formation (Appendix A), supporting the hypothesis that Mdm4 tumorigenic activity is dependent on the amount of Mdm4, in agreement with a previous report [23]. Analysis of tumor growth did not show a significant difference between Mdm4^Tg15^ and WT animals both in male and female groups, although transgenic mice show a tendency of increased growth rate (Figure 1D,E). These data suggest that Mdm4 oncogenic properties are mainly operating in the initial phases of tumor development. Protein analysis confirmed Mdm4 overexpression both in tumors and contra-lateral healthy muscles from Mdm4^Tg15^ mice, with high variability among animals (Figure 1F) whereas endogenous Mdm4 was undetectable in the muscle of WT animals (Figure 1F). Overall, these data indicate that Mdm4 accelerates DNA-damage induced tumor formation without being affected by mouse gender.

Mdm4 is mainly known as p53-regulator, but it also shows in vivo p53-independent oncogenic properties [14]. To uncover whether Mdm4 accelerates fibrosarcoma formation by suppressing p53 function, we sequenced p53 alleles in a subset of 22 tumors, deriving from WT and transgenic animals of both genders. More than half of the tumors (12/22) have mutations of at least one p53 allele independently of the genotype (WT or Mdm4^Tg^) and the gender of the animals (Table 1 and Appendix A). Therefore, overexpression of Mdm4 did not reduce the selective pressure to inactivate p53 in these tumors.

### 2.2. Cisplatin-Sensitivity Is Affected by Mdm4 in a Gender-Dependent Manner

Since MDM4 exerts a pro-apoptotic activity under stress conditions [16,25] and its high levels have been associated with increased sensitivity to DNA damage and cisplatin treatment [19,26], we tested chemosensitivity of 3MCA-induced fibrosarcoma in WT and Mdm4^Tg^ mice. After reaching a tumor growth of approximately 200 mm^3^, mice were treated bi-weekly with cisplatin, and tumor size was monitored. Tumors were sensitive to cisplatin as indicated by reduced volume compared to the exponential growth of untreated tumors (compare Figure 2A,B to Figure 1D,E). After an initial period of a similar response to cisplatin treatment, fibrosarcoma from Mdm4^Tg15^ males maintained an almost complete growth suppression compared to WT animals (Figure 2A), a phenomenon evident also in Mdm4^Tg10^ line (Appendix A). Accordingly, 8/10 Mdm4^Tg15^ mice displayed complete tumor remission compared to 1/7 WT animals, with a significantly higher tumor-free survival period (Figure 2C).

Interestingly, these results were not observed in females (Figure 2B,D and Appendix A), in which tumor growth showed similar behavior, although with reduced kinetics in Mdm4^Tg^ mice. Further, the number of females displaying complete tumor remission was similar (1/7 for Mdm4^Tg15^ and 1/8 for WT) with identical relapse time (17 days) (Figure 2D). Analysis of p53 mutation in a subset of tumors from three MDM4^Tg^ and WT mice showed similar frequency of mutation with no overt correlation with tumor growth rate, maybe due to their appearance at different times along tumor evolution. These data are the first evidence in vivo that Mdm4 enhances tumor chemosensitivity, thus confirming its “anticancer” activity under stress conditions. Of note, this activity shows sexual dimorphism, with males only displaying susceptibility to Mdm4 levels compared to females.

Chemotherapy sensitivity in vivo is related to many factors, with cancer-associated environment highly contributing to this sensitivity [27]. To ascertain whether the observed sexual dimorphism is cell-autonomous and specific to cisplatin, we analyzed cell death of thymocytes, a cell type that expresses the transgene Mdm4 (Appendix A) and is highly sensitive to DNA damage. Following sub-lethal total body ɣ-irradiation, thymocytes of Mdm4^Tg^ males showed decreased cell viability whereas those from Mdm4^Tg^ females did not show different survival compared to WT mice (Figure 2E,F). These data confirm that: (i) Mdm4-mediated sensitivity to acute DNA damage is gender-specific; (ii) this alteration is at least in part cell-autonomous; (iii) it occurs in primary cells too, and therefore it is not specific to the cancer background. 

To investigate whether estrogens, the most specific female hormones, are responsible for this phenomenon, tumor-bearing mice were treated with cisplatin in absence or presence of the estrogen receptor-inhibitor fulvestrant (ICI 182,780), a selective estrogen receptor degrader (SERD). Interestingly, ICI treatment reduced tumor growth significantly in Mdm4^Tg15^ females (Figure 2G), indicating an antagonistic function of estrogen receptors towards Mdm4-mediated cisplatin sensitivity. Conversely, ICI was ineffective in WT mice (Figure 2H), in agreement with the undetectable presence of Mdm4 in mouse muscle (see Figure 1C,F). Therefore, the sexual dimorphism observed in Mdm4-mediated chemosensitivity is mediated by the estrogen pathway. As control of ICI treatment, estrogen-transcriptional targets progesterone receptor—PgR-and insulin-like growth factor—IGF-1—were significantly reduced in hormone-sensitive tissues breast and vagina both in transgenic and WT mice (Appendix A), also indicating that estrogens are functioning in both groups.

### 2.3. Estrogens Modulate Mdm4 Intracellular Localization 

To understand the mechanism by which estrogens modulate Mdm4 activity, we used the previously described in vivo model and analyzed the mRNA levels of genes involved in thymocyte apoptosis following ɣ-irradiation in male and female WT and Mdm4^Tg^ mice [28,29]. RT-PCR analyses showed strong upregulation of the expression of apoptosis-related genes *Bax*, *PMAIP1/Noxa*, *BBC3/Puma*, *p21/Waf1*, and *LGALS3* upon ɣ-irradiation (Appendix A). However, no significant difference was observed between WT and transgenic animals (Appendix A), indicating that Mdm4 overexpression does not alter the transcriptional profile of these genes, many of which are p53-targets. We, therefore, tested in vitro models of estrogen-sensitive (MCF-7 and T-47D) and ERα-negative (MDA-MB-231 and A2780) breast and ovarian cancer cell lines with wt (MCF-7, A2780) or mutant p53 (T-47D, MDA-MB-231). 17β-estradiol (E_2_) was indeed able to counteract cisplatin-induced cell death in MCF-7 (Figure 3A), and T-47D cells (Appendix A), and ICI treatment counteracted E_2_ activity (Figure 3A), confirming the ERα-mediated pro-survival function of E_2_ in these human cell lines, independently of p53 status. Conversely, E_2_ did not affect cell proliferation nor cisplatin-sensitivity of triple-negative MDA-MB-231 cells (Figure 3B), confirming the requirement of ERα for hormone activity. Under these conditions, MDM4 levels were not substantially altered by E_2_ in MCF-7 cells, independently of cisplatin treatment (Figure 3C and Appendix A). On the contrary, as control, levels of p53 and its target MDM2 were highly induced by cisplatin (Figure 3C). To proof ERα activity, the mRNA levels of PgR and pS2/TFF1, two ERα target genes, were strongly induced by E_2_ treatment (Appendix A) and the levels of the receptor were decreased (Figure 3C), as negative feedback of estrogen receptor stimulation by its ligand in estrogen-responsive tissues. Accordingly, the levels of nuclear ERα relative to the total levels were increased (Appendix A) as a result of increased nuclear activity.

MDM4 is mainly cytoplasmic in untransformed cells [30], and cytoplasmic localization has been associated with its pro-apoptotic activity [19,31]. Conversely, upon DNA damage, it shifts into the nucleus [32]. Indeed, we observed a decrease of MDM4 cytoplasmic levels and increase of nuclear ones in MCF-7 cells upon cisplatin treatment (Figure 3D,E). Interestingly, the presence of estrogens significantly increased the nuclear fraction of MDM4 after cisplatin treatment (Figure 3D,E). This effect was also observed in T-47D cells even in the absence of cisplatin (Appendix A) whereas it did not occur in ERα-negative MDA-MB-231 cells (Appendix A). Thus, ERα can alter MDM4 pro-apoptotic function by altering its intracellular localization. In agreement with the reported MDM4 degradation following its nuclear localization [33], a time-course analysis revealed decreased levels of MDM4 in the presence of estrogens at later time points (Appendix A)

To confirm the estrogen-dependent nuclear re-localization of MDM4, we performed co-immunofluorescence analysis of ERα and MDM4 in A2780 ovarian cancer cells. The two proteins were transiently overexpressed and their subcellular localization analysed. MDM4 was mainly cytoplasmic independently of the co-expression of ERα or sole E_2_ treatment (Figure 3F,H,I).

Conversely, the number of cells showing nuclear/cytoplasmic or exclusively nuclear MDM4 was strongly increased by E_2_ when MDM4 and ERα were co-expressed in the same cell (Figure 3G,I). ERα was mainly nuclear and E2 treatment led to its almost complete nuclear localization. MDM4 expression did not alter this behavior (Appendix A). These data confirm that 17β-estradiol through its receptor alpha is responsible for shifting MDM4 into the nucleus. Since MCF7 express a fraction of MDM4 into the nucleus (Figure 3D) [19]—a feature frequently occurring in human tumor cells [34]—it was not possible to quantitatively analyze MDM4 re-localization by immunofluorescence upon estradiol treatment in these cells. Since the machinery responsible for MDM4 nuclear localization has not been identified, we investigated the interaction of MDM4 and ERα under different growth conditions. The lack of interaction between the two proteins in our system excludes re-localization of MDM4 by ERα through their interaction (Appendix A).

Given the lack of available antibodies to detect murine Mdm4 by IHC in fibrosarcoma, to ascertain whether this mechanism can occur in vivo, subcellular localization of Mdm4 was analyzed in thymocytes of Mdm4^Tg^ mice. Since the levels of Mdm4 are rapidly disappearing in thymocytes as soon as after 4 h from DNA-damage, we analyzed Mdm4 levels in the absence of DNA damage. To boost the effect of E_2_, transgenic females were fed with a diet containing reduced levels of estrogen-like substances and after 5 days treated with a single acute i.p. injection of the hormone. After 16 h, animals were sacrificed, and thymocytes isolated for subcellular fraction analysis. As previously reported, ERα is expressed in these cells [35], and its levels decrease consequently to E_2_ treatment (Figure 4A). In E_2_-treated transgenic mice, total levels of Mdm4 were significantly decreased compared to vehicle-treated animals whereas the levels of its homolog Mdm2 were not similarly altered (Figure 4A). Of interest, the Mdm4 decrease was associated with increased nuclear levels correlated to a decrease of the cytoplasmic ones (Figure 4B,C). These data indicate that 17β-estradiol re-localizes Mdm4 also in vivo, supporting and further expanding our previous findings. In primary cells, this is accompanied by a rapid decrease of Mdm4 levels. Overall, these data support a model whereby E_2_/ERα-mediated re-localization and decrease of MDM4 levels, antagonize its activity in the female gender.

### 2.4. Mdm4 Intracellular Localization Correlates with Platinum Sensitivity in High-Grade Serous Ovarian Carcinoma

Estrogens are an important factor in ovarian cancer with intra-tumor production of estrogens being considered one of the main sources of these hormones [36,37]. Although increased MDM4 levels have been associated with better prognosis and cisplatin chemosensitivity in ovarian cancer [19,26], subcellular localization of MDM4 has not been investigated in tissue specimens. To this aim, MDM4 expression and subcellular localization were retrospectively analyzed in a group of high grade serous ovarian carcinomas (HGSOC) (Appendix A). To avoid bias from the endogenous menstrual cycle, we considered cancer samples from post-menopausal women. Majority of patients underwent platinum/taxol chemotherapy and were defined as resistant or sensitive according to Colombo et al. [38]. Tumor cells were classified as presenting only nuclear or nuclear and cytoplasmic MDM4 signal, and a final score (based on the intensity and the percentage of positive cells) was assigned to each tumor, for each pattern of MDM4 expression. A variable subcellular localization of MDM4 was indeed observed in these samples (Figure 5A). Of interest, a significantly increased proportion of cells presenting exclusively nuclear MDM4 was observed in the chemotherapy-resistant group compared to the sensitive one (Figure 5B) whereas the overall levels of MDM4 were not significantly different among two groups (Appendix A). The high number of ovarian cancer samples showing exclusive-nuclear staining compared to A2780 cells overexpressing MDM4 is probably due to the different levels of MDM4 in the two cell systems. To ascertain whether the estrogenic pathway is correlated to this localization, we analyzed the levels of ERα. The receptor was expressed in all specimens with no significant difference between resistant and sensitive patients (6.9 ± 3.5 and 6.5 ± 3.7, mean ± SD respectively), in agreement with the high proportion of ovarian tumors expressing ERα [39]. Since these samples derive from post-menopausal women, to ascertain whether ERα is activated by its ligand/s, we analyzed the levels of steroid sulfatase (STS). STS plays an essential role in the conversion of inactive biological steroids to active E_2_, especially in post-menopausal women [40,41]. Its activity has been correlated with E_2_ serum levels [36], and its low levels have been associated with longer progression-free survival in ovarian cancer [42,43]. The mRNA levels of STS in available matched frozen samples were indeed higher in the resistant compared to the sensitive group (Figure 5C), indicating increased local availability of active estrogens in the first group. These data support the hypothesis that increased estrogen-mediated activity of ERα in these tumors mediates the observed nuclear localization of MDM4.

To have further insight into the activity of nuclear MDM4 in the chemo-resistance of HGSOC samples, active caspase-3 (CC3) was analyzed as a sign of cell death. As expected, CC3 was significantly reduced in the resistant group compared to the sensitive one (Figure 6A,B). Interestingly, an inverse relationship was observed between nuclear MDM4 and CC3 (*p* < 0.0001, r = 0.7) (Figure 6C), supporting the anti-apoptotic activity of nuclear MDM4 and its relevance in the chemo-resistance of these samples. Importantly, nuclear MDM4 expression correlated with disease-free (DFS) survival while approaching statistical significance with overall survival (OS) (*p*< 0.05 and *p* = 0.06, respectively, Figure 6D,E). Noteworthy, a strong difference in median values was observed. Indeed, median DFS values were 25 and 12 months in low and high nuclear MDM4 patients, respectively with the hazards ratio (HR) of low versus high MDM4 expression equal to 0.4 (95% CI 0.2–1.0) (Figure 6D). Likewise, median OS values were 85 and 38 months in patients with low and high nuclear MDM4 expression (HR 0.4; 95% CI 0.2–1.0) (Figure 6E). 

These data confirm the association between nuclear MDM4/estrogens and chemosensitivity in HGSOC and suggest a novel pathophysiological function of estrogens in this cancer.

## 3. Discussion

In this work, we investigated whether in vivo estrogenic hormones affect the functions of MDM4, by using an in vivo model of transgenic mice overexpressing the murine Mdm4 protein in the C57Bl/6J background.

We observed that high levels of murine Mdm4 accelerate the formation of DNA-damage induced fibrosarcoma in a gender-independent manner. Conversely, the transgene increases the sensitivity of these tumors to cisplatin in a gender-dependent manner with females being less sensitive to the increased levels of Mdm4. These data point to two important MDM4 in vivo properties: (1) MDM4 possesses a dual mode of action. It shows oncogenic properties in DNA-damage induced tumorigenesis—that can be considered a long-term DNA-damage response—while anticancer properties in acute chemotherapy, a short term DNA-damage response; (2) the gender affects the sole ability of MDM4 to enhance acute DNA damage-mediated cell death. These opposite consequences of MDM4 activity in response to two conditions of DNA damage (3MCA or Cisplatin) suggest that MDM4-mediated cell death poorly contributes to suppressing cell transformation at least in fibrosarcoma whereas it is relevant in response to chemotherapy. These data are in agreement with previous studies which excluded a prominent role of cell death in antagonizing cell transformation: efficient p53-mediated tumor suppression requires the activation of Arf by oncogenic pathways and not the acute DNA damage response [44,45]. An additional factor could be the occurrence of MDM4-oncogenic activities towards other oncosuppressor pathways, besides the suppression of p53. Although the high frequency of p53 mutation observed in fibrosarcoma might suggest a p53-independent function of MDM4, this event could have occurred later in established tumors, and MDM4 could have acted on p53 in the first phases of tumor development. This last hypothesis is supported by the evidence that tumor growth is not significantly affected by the overexpression of Mdm4.

The use of ICI/fulvestrant allowed us to define the relevance of estrogens in the sexual dimorphism of Mdm4-mediated chemo-sensitization. Given the high affinity of this inhibitor towards ERα, these data prove the antagonizing function of the estrogenic pathway towards MDM4. In agreement with our data, a recent study reported an inverse relationship between estrogen receptor status and tumor response in breast cancer patients treated with neoadjuvant chemotherapy [46]. To note, breast cancer is a tumor with a high incidence of MDM4 overexpression [10]. Based on these and our results, we think that detailing the estrogen receptor status and the local availability of hormones in tumors expressing MDM4, could validate the predictive value of these parameters and guide optimal therapeutic combination. 

In vitro studies on primary cells and tumor cell lines gave insight into the molecular mechanism by demonstrating that ERα alters the subcellular localization of both mouse and human MDM4, increasing its nuclear fraction and subsequent degradation. The increased degradation of MDM4, observed especially in thymocytes, contributes to explaining the prognostic value of MDM4 SNP34091 and levels reported solely in ER-negative cancer. Indeed, the ability of estrogens to control MDM4 degradation probably cancels the differences in MDM4 expression. However, the sole reduction of MDM4 levels is not sufficient to explain the antagonistic function of ERα, also if considering the similar total levels observed in HGSOC samples. Since the pro-apoptotic activity of MDM4 has been associated exclusively to its cytoplasmic localization, the ability of ERα to increase MDM4 nuclear fraction also antagonizes its cytoplasmic functions. In agreement with this model, the majority of tumors analyzed thus far show nuclear localization of MDM4 [34]. These data also suggest a possible bias in interpreting data deriving from cell culture since cell culture medium is usually rich in estrogens and estrogen-like substances that may alter MDM4 activities.

At present, the shuttling system responsible for MDM4 nuclear localization has not been identified. Our data exclude a direct MDM4/ERα interaction, whereas they suggest the involvement of ERα transcriptional function. In this respect, they contradict the work of Swetzig et al., [47] who showed the interaction between the two proteins. Different receptor manipulation (estrogen-stimulated versus genetic ablation) may underlie this discrepancy. Of note, the ability of estrogens to modulate cyto-nuclear shuttling has been highlighted by a recent proteome analysis in MCF-7 cells. This study evidenced that dynamic subcellular redistribution of many proteins in response to estrogens is the major phenomenon compared to the alteration of protein levels [48]. Two studies have further indicated the ability of ERs to re-localize cellular factors [49,50] supporting a novel way of action for these hormones. Interestingly, a recent perspective introduced the relevant role of STRaNDs (shuttling transcriptional regulators and non-DNA binding), proteins that modify cell response/transcription in response to external stimuli by migrating into the nucleus without binding to DNA [51]. Besides including MDM4 among STRanND factors, our data support the importance of cyto-nuclear shuttling in cell response and pathophysiology and include estrogens among mastermind of STRanND proteins. 

Of relevance for the clinic, the subset of analyzed HGSOC samples supports the estrogen function indicated by in vivo and in vitro data. Although the reduced number of patients, our results suggest that the local availability of estrogens may be an important factor in the pathophysiology of this cancer and an element to consider for treatment. Moreover, the great differences in median values of DFS and OS highly support the possible predictive value of MDM4 localization.

Overall, these results suggest a novel pathogenetic mechanism of estrogens that can be relevant in the application of radio- and chemotherapy to tumors expressing MDM4. Furthermore, given the proximity of MDM2/MDM4 inhibitors to clinical application, our data may be useful to define the clinically relevant subgroups and to investigate optimal combination schedule for these drugs.

## 4. Materials and Methods

### 4.1. Mouse Maintenance and Treatment

Control (WT) and Mdm4 transgenic (MDM4^Tg^) mice were maintained and treated in accordance with the Guidelines on the protection of animals used for scientific purposes (European Directive 63/2010/EU and Italian Law DL116/1992 and DL 26/2014). Relative ethical approval has been obtained by Animal Welfare Body “Fondazione S. Lucia” (Protocol Number:16/2014). All experiments were performed in animals back-crossed to C57Bl/6J to generate Mdm4^Tg^ mice that were >93% C57Bl/6J background. For fibrosarcoma development, 3-methylcholanthrene (0.5 mg/100ul) was inoculated in the hindlimb muscle. After about 3 months, the tumor mass became palpable. Mice were examined twice a week for tumor development and size, using caliper at the indicated time points. Tumor size was calculated using the following formula: V = (W(2)xL)/2 where W is the tumor width and L is the tumor length. When tumors reached a volume of approximately 200 mm^3^, mice were treated with ciplatin ± ICI. The cisplatin (TEVA, Petah Tiqwa, Israel) dose was 5 mg/kg inoculated i.p. twice a week. ICI (Faslodex, Astra Zeneca, Cambridge, UK) dose was 200 mg/kg inoculated s.c. weekly. The animals were euthanized for ethical reasons when the tumor reached a weight of about 10% of mouse weight or when animals showed signs of sufferance. Euthanasia occurred by CO_2_ at saturation levels greater than 70%. For ɣ-irradiation experiments, age-paired WT and Mdm4^Tg^ mice received a sublethal dose of 6Gy using a cesium-137 irradiator. After 6 h, mice were sacrificed by CO_2_, the thymus was removed and thymocytes isolated by filtration using a 100 µm mesh membrane (Millipore, Burlington, MA, USA). For the experiments in Figure 4, Mdm4^Tg^ females were fed with Teklad global 2016 (Envigo, Huntingdon, UK). After 5 days, mice were i.p. injected with 50 µg/Kg of 17β-estradiol (Faslodex) or physiologic solution and euthanized after 16 h for thymocytes isolation and subcellular fraction analysis. 

### 4.2. Cell Cultures and Treatments

Human breast cancer cell lines, MCF-7, T-47D, and MDA-MB-231 were maintained in DMEM, human ovarian cancer cell line A2780 in RPMI medium, all containing fetal bovine serum 10%, 1% stable glutamine and 1% penicillin-streptomycin (Invitrogen, Carlsbad, CA, USA), and kept at 37 °C in 5% CO2. Cell line identity has been confirmed by short tandem repeat (STR) profiling (Eurofins, Luxembourg, Luxembourg, last analysis 2019). Mycoplasma-free conditions have been routinely tested by MycoAlert kit (Lonza, Basel, Switzerland). Transient transfection in A2780 cells was performed using Lipofectamine Plus (ThermoFisher, Waltham, MA, USA). For cell viability and cell death assays, 5 × 10^3^ cells were seeded in a 96 multiwell in media deprived of estrogen-like compounds (DMEM w/o phenol red, with 10% Charcoal dextran treated FBS, 1% stable glutamine and 1% penicillin-streptomycin) and grown for 48 h. Cells were then treated with 1 nM 17β-estradiol ± ICI for 24 h and treated with 40 μM cisplatin ± ICI /17β-estradiol for 72 h. Cell death and viability luminescence emission were evaluated using Enspire device (PerkinElmer, Waltham, MA, USA).

For nuclear/cytosolic fractionation, cells were seeded in a 100 mm plate in media deprived of estrogen-like compounds and grown for 48 h. Cells were then treated with 1 nM 17β-estradiol for 8 h and treated with 40µM cisplatin for 18 h. For thymocytes analysis, thymi were isolated from irradiated mice and homogenized in PBS. Primary cells were isolated using a 100 mesh filter, and RBC lysis was performed using a hypotonic solution (Sigma, St. Louis, MO, USA). Samples were stained with PI and immediately analyzed by a FACScan cytofluorimeter (BD, Franklin Lakes, NJ, USA) using the BD CellQuest software package.

### 4.3. Immunofluorescence

A2780 cells were plated in media deprived of estrogen-like compounds and grown for 48 h. Cells were then transfected with pcDNA3.1hMDM4 and pHEGO ERα plasmids and treated with 1 nM 17-β-estradiol or ethanol. Cells were then fixed with 3.7% formaldehyde in PBS, permeabilized with 0,05% Triton-X100 and unspecific binding saturated with 5%BSA. The following primary antibodies were used: α-MDM4 clone 4B5 1:100 (#TA505706, Origene, Rockville, MD, USA), α-ERα HC-20 1:100 (#sc-543 Santa Cruz Biotechnology Inc., Dallas, TX, USA). Donkey α-mouse Cy3 and donkey α-rabbit Cy2 (Jackson ImmunoResearch, Cambridge, UK) were used as secondary antibody and cells were mounted with ProLong/DAPI (ThermoFisher, Waltham, MA, USA). Images were taken using an SP5 confocal fluorescence microscope (Leica, Wetzlar, Germany).

### 4.4. Protein Analysis

Nuclear and cytoplasmic fractions of cells were prepared as follows: cells scraped with PBS were resuspended in hypotonic lysis buffer (10 mM HEPES pH 7.9, 10 mMKCl, 0.1 mM EDTA, 0.1 mM EGTA) added with protease inhibitors (Roche, Basel, Switzerland) and incubated on ice for 15 min. After resuspension, NP-40 was added to a final concentration of 0.6%, and the nuclei were pelleted by centrifugation at 10,000 g for 30 s at 4 °C. The supernatant was collected as cytoplasmic fraction. Nuclei pellets were resuspended in nuclear extraction buffer (20 mM HEPES pH 7.9, 25% glycerol, 0.4 M NaCl, 0.1 M EDTA, 0.1 mM EGTA) and sonicated by a Sonicator W-375 Cell Disruptor (Heat Systems Ultrasonics Inc., Newtown, CT, USA) at 20% of maximum output power for 30 s. After rocking for 15 min at 4°C nuclei extracts were recovered as supernatant by centrifugation at 14,000 g for 15 min at 4 °C. For WCEs, cells were lysed with RIPA buffer (50 mMTris–Cl, pH 7.5, 150 mMNaCl, 1% NP-40, 0.5% Na deoxycholate, 0.1% SDS, 1 mM EDTA) supplemented with a cocktail of protease inhibitors (Roche, Basel, Switzerland).

For Western blot, all SDS–PAGE were transferred onto PVDF membranes (Millipore, Burlington, MA, USA). Membranes were developed using the enhanced chemiluminescence (ECL Amersham, Little Chalfont, UK and Cyanagen, Bologna, Italy) by the chemiluminescence imaging system Alliance 2.7 (UVitec Cambridge, UK) and quantified by the software Alliance V_1607.

The following primary antibodies were used: α-human MDM4 clone 8C6 1:1000 (#04-1555 Millipore, Burlington, MA, USA), α-mouse Mdm4 1:1000 (#M0445, Sigma), mix of α-MDM2 clone SMP14 1:1000 (sc-965, Santa Cruz Biotechnology Inc.,) and 2A10 1:50 (kindly provided by M.E. Perry) for murine Mdm2, mix of clone Ab1 1:1000 (#OP46, Millipore) and 2A10 1:50 for human MDM2, α-ERα HC-20 1:1000 (#sc-543 Santa Cruz Biotechnology Inc.), α-ERα MC-20 1:1000 (#sc-542 Santa Cruz Biotechnology Inc.), anti-p53 FL-393 1:1000 (#sc- 6243 Santa Cruz Biotechnology Inc.), α-Lamin B C-20 1:1000 (#sc6216 Santa Cruz Biotechnology Inc.), α-GAPDH 1:8000 (#GA1R, ThermoFisher), α-actin C-40 1:4000 (#A4700, Sigma), α-Tubulin DM1A 1:4000 (#T9026, Sigma), α-PCNA PC10 1:1000 (#SC-56, Santa Cruz Biotechnology Inc.), α-ERβ N2C2 1:1000 (#GTX110607, GeneTex, Irvine, CA, USA), horse-radish peroxidase-conjugated anti-rabbit 1:8000 (#Sc-2054 Santa Cruz Biotechnology Inc.), HRP α-goat 1:8000 (#Sc-2768 Santa Cruz Biotechnology Inc.), HRP α-mouse 1:5000 (Bio-Rad, Hercules CA, USA). 

### 4.5. Quantitative Real-Time Polymerase Chain Reaction (qPCR) 

Total RNA was isolated using Trizol Reagent (ThermoFisher, Newtown, CT, USA) following the manufacturer’s instructions. After in vitro retrotranscription, Real-Time PCR was performed using an ABI 7300 Real-Time PCR System with 7300 System SDS Software (ThermoFisher, Newtown, CT, USA) using SensiMix SYBR Hi-ROX Mix (Bioline, London, UK). The detection of a single amplicon was verified using a dissociation curve. Normalized, relative mRNA levels were calculated according to the ΔΔ*C*t method, using endogenous reference gene for normalization. Primers efficiencies were calculated from a dilution curve and determined to be within the acceptable range of 90–110% efficiency.

The following primer sets were used for qPCR: mouse PgR5′- GGTGGAGGTCGTACAAGCAT-3′ and 5′- GCTCCTTCATCCTCTGCTCAT-3′; mouse Igf-1 (5′-CTACAAAAGCAGCCCGCTCTA-3′ and 5′- TAGGGACTTCTGAGTCTTGG-3′; mouse β-Actin 5′-CGA TGC CCT GAG GCT CTT T-3′ and 5′-TAGTTTCATGCATGCCACAGGAT-3′, mouse TBP 5′CCAATGACTCCTATGACCCCTA-3′ and 5′-CAGGCAAGATTCACGGTAGAT-3′; mouse Mdm4 5′-TCGCACAGGATCACACTATGG-3′ and 5′-TTATGTCGTGAGGTAGGCAGT-3′; mouse NOXA 5′-CAGATGCCTGGGAAGTCG-3′ and 5′-TGAGCACACTCGTCCTTCAA-3′; mouse p21 5′-GCAGATCCACAGCGATATCCA-3’ and 5’-AGACAACGGCACACTTTGCTC-3’; mouse BAX 5′-GTTTCATCCAGGATCGAGCAG-3′ and 5′-CCCCAGT TGAAGTTGCCATC-3′; mouse PUMA 5′-TTCTCCGGAGTGTTCATGC-3′ and 5′-GATACAGCGGA GGGCATC-3′; mouse LGALS3 5′-GCCTACCCCAGTCTCCT and 5′-GGTCATAGGGCACCGTCA; human PgR 5′- TGGAAGAAATGACTGCATCG-3′ and 5′-TAGGGCTTGGCTTTCATTTG; human Ps2 5′-TTGTGGTTTTCCTGGTGTC-3′ and 5′-CCGAGCTCTGGGACTAATCA-3′; human MDM4 5′-TCGCACAGGATCACAGTATGG -3′ and 5′-CAGTGTGGGGATATCGTCTTTCT; human GAPDH 5′-GAGTCAACGGATTTGGTCGT-3′ and 5′-GACAAGCTTCCCGTTCTCAG-3′. Human STS was #69919842 from IDT (Leuven, Belgium).

### 4.6. Patients

The study included 33 patients with advanced high grade serous ovarian cancer admitted to the Gynecologic Oncology Unit, Fondazione Policlinico Universitario A. Gemelli, between January 2002 and December 2008. Clinicopathological characteristics of the overall series are summarized in Appendix A. Recurrence of disease was defined according to GCIG (Gynaecological Cancer Intergroup) CA125 criteria [38,52] and/or radiological confirmation of tumor progression.

Chemosensitivity was defined according to the common definition of platinum resistance [38], identifying as “sensitive” patients those that relapsed 6 months or more after prior platinum-containing chemotherapy, and as “resistant” patients those that relapsed less than 6 months after chemotherapy was stopped, or that progressed during therapy. Follow-up data were available for all 33 patients (median follow-up, 43 months; range, 9–140 months). During the follow-up period, progression and death of disease were observed in 26 and 21 patients, respectively. The trial was approved by the local Ethics Committee and Institutional Review Board (Protocol: A.2019/CE/2012). All patients signed a written informed consent by agreeing to submit to all the procedures described and for their data to be collected.

### 4.7. Immunohistochemistry

Three-micrometer-thick whole sections from the formalin-fixed, paraffin-embedded tissues were mounted on Superfrost coated slides, and dried overnight. The sections were deparaffinized in xylene and rehydrated in graded solutions of ethanol; the endogenous peroxidase was blocked with 3% H_2_O_2_ for 5 min. Immunohistochemistry for MDM4 (A300-287A-2, dilution 1:100, Bethyl Laboratory, Montgomery, TX, USA), cleaved caspase-3 (CC3) (Cell Signaling Technology, Leiden, The Netherlands, clone 5A1E, dilution 1:100) and ERα (clone 6F11, Biocare, Concord, CA, USA, dilution 1:100) were performed using a labeled streptavidin-biotin peroxidase method on individual sections from each tumor. Antigen retrieval procedure was performed by microwave oven heating in citrate buffer (pH = 6) for MDM4 and CC3, and EDTA (pH = 9) for ERα. Cells expressing MDM4 were identified after 25 min incubation at room temperature, while cell expressing CC3 or ERα were identified after overnight incubation at 4 °C. Slides were developed with diaminobenzidine (DAB substrate System, Dako Cytomation, Santa Clara, CA, USA), counterstained with Mayer’s Haematoxylin, dehydrated in ethanol and xylene, and finally mounted. Scoring of ERα and subcellular localization of MDM4 was evaluated as previously reported [52]. Briefly, mean percentage of stained cells (both nuclear and cytoplasmic) was categorized as follows: 0 = 0%, 1 = 1–10%, 2 = 11–33%, 3 = 34–66% and 4 = 67–100%. The staining intensity was also evaluated and graded from 0 to 3, where 0 = no staining, 1 = weak staining, 2 = moderate staining, and 3 = strong staining. The two values obtained were multiplied to calculate a receptor score (maximum value 12). For outcome analysis, patients expressing MDM4 were grouped into low (score below or equal to 2) or high (score higher than 2) nuclear expression. Expression of CC3 was evaluated by considering the number of cells exhibiting immunoreaction in a minimum of 500 histologically identified neoplastic cells; data are expressed as a percentage of marker-positive cells among total cells.

### 4.8. Statistics 

Data are reported as means ± SD, and the indicated tests were used for the determination of the statistical difference between groups. DFS was defined as the interval (in months) between the date of diagnosis and date of progression (radiological or clinical assessed) or death whichever occurred first, or date of the last follow-up for patients alive and without progression. OS was defined as the interval between the date of diagnosis and date of death (or date of the last follow-up for alive patients). The prognostic effect of MDM4 on clinical outcome (i.e., recurrence of disease or death) was tested by plotting survival curves according to the Kaplan–Meier method, and comparing groups using the Log-rank test. *p* values were two-sided, with *p* < 0.05 considered as significant. Statistical Analysis was performed using GraphPad Prism 5 (GraphPad Software Inc., San Diego, CA, USA).

## 5. Conclusions

In this study, we addressed the impact of estrogenic hormones on the function of Mdm4, a protein overexpressed in many tumors and implicated in the p53 pathway. We demonstrated that the estrogens are able to alter Mdm4 activities by modifying its intracellular localization. Particularly, estrogens increase nuclear localization and consequent degradation of Mdm4, thus antagonizing its cytoplasmic pro-apoptotic function. As a result, female mice bearing Mdm4-overexpressing fibrosarcoma are less sensitive to cisplatin treatment compared to male mice.

Combination therapy of female mice with fulvestran and cisplatin is able to recover chemotherapy sensitivity, thus suggesting the potential beneficial effect of antiestrogenic therapy in Mdm4-expressing tumors. To support and expand these findings to human beings, a close relationship was further demonstrated in HGSOC, a tumor sensitive to Mdm4 levels, among MDM4 nuclear localization, apoptotic sensitivity to platinum-based therapy and estrogen levels, suggesting the possible predictive value of MDM4 localization for HGSOC chemosensitivity.

## Figures and Tables

**Figure 1 cancers-11-01349-f001:**
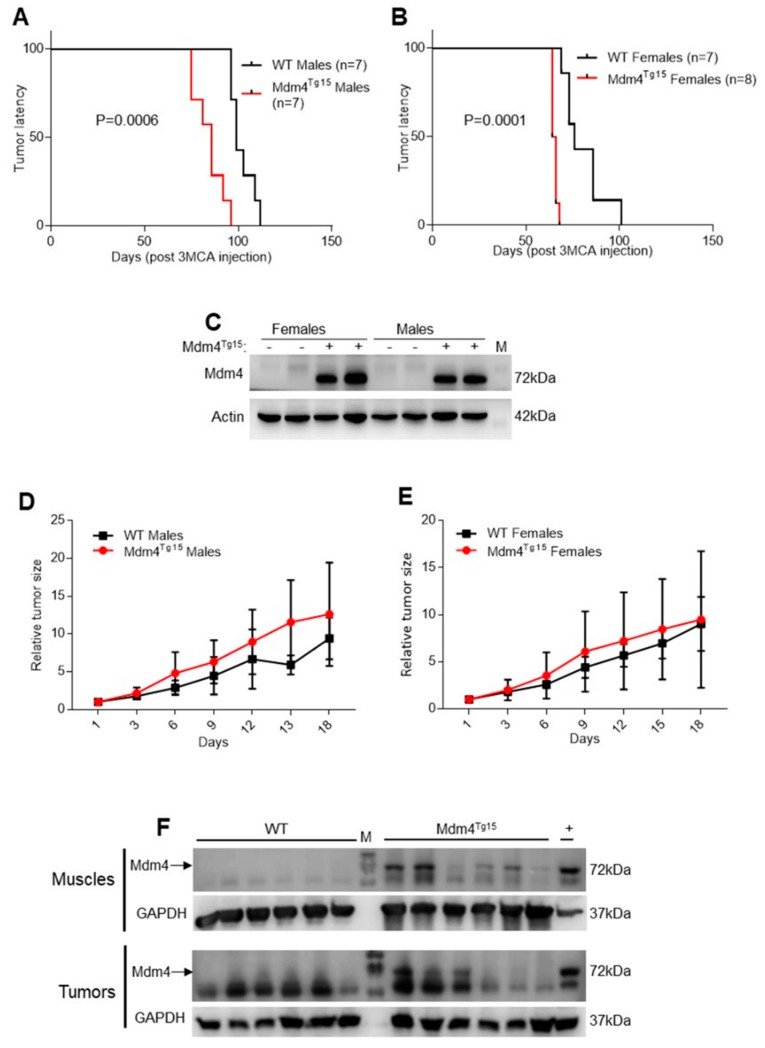
Mdm4 accelerates DNA-damage induced tumorigenesis. (**A**,**B**) Tumor latency in wildtype (WT) and transgenic Mdm4^Tg15^ males (**A**) and females (**B**). Fibrosarcomas were induced by a single injection of 3-MCA in the hindlimb muscle of age-matched animals (Log-rank test (**A**) df = 1, χ^2^ = 11.87; (**B**) df = 1, χ^2^ = 14.45). The data are representative of 3 (**A**) and 2 (**B**) independent experiments. (**C**) Western blot (WB) analysis of hindlimb muscle from age-matched WT and Mdm4^Tg15^ males and females. (**D**,**E**) Growth curve of fibrosarcoma in WT and Mdm4^Tg15^males (**D**) and females (**E**). Tumor size is relative to the size at the starting point (two-way ANOVA, (**D**) DF = 6, F_(interaction)_ = 1.162, *p* = 0.338; (**E**) DF = 6, F_(interaction)_ = 0.083, *p* = 0.9975). (**F**) WB analysis of Mdm4 levels in tumors and matched muscles derived from animals reported in panel (**A**). GAPDH was used as a loading control (LC). M indicates molecular weight.

**Figure 2 cancers-11-01349-f002:**
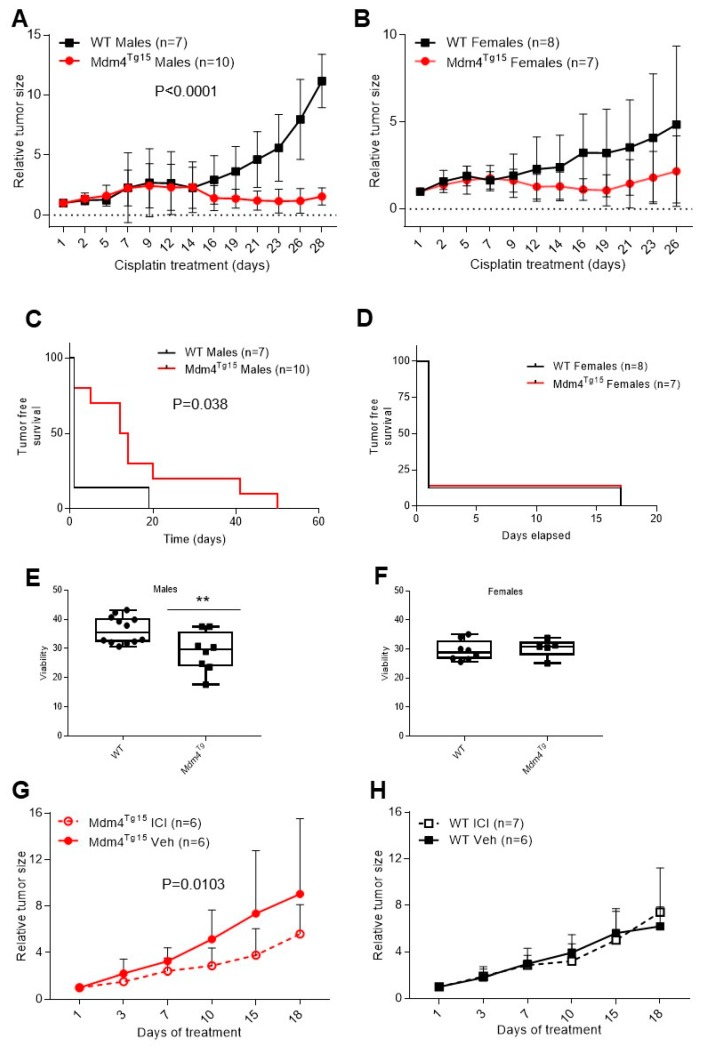
Cisplatin-sensitivity is decreased in transgenic Mdm4^Tg15^ females, dependent of the ERα activity. (**A**,**B**) Growth curve of fibrosarcoma in WT and Mdm4^Tg15^males (**A**) and females (**B**) treated with cisplatin. After reaching a volume of approximately 200 mm^3^, animals were treated bi-weekly with i.p. cisplatin (5 mg/Kg). Tumor size is relative to the volume at the first treatment (two-way ANOVA, (**A**) DF = 12, F_(interaction)_ = 9.109; (**B**) DF = 11, F_(interaction)_ = 1.194). (**C**,**D**) Tumor-free survival of Mdm4^Tg15^ and WT males (**C**) and females (**D**) shown in A and B (Log-rank test (**C**) df = 1, χ^2^ = 4.27; (**D**) df = 2, χ^2^ = 3031, n.s.). (**E**,**F**) Thymocyte viability in Mdm4^Tg^ and WT males (**E**) and females (**F**) subjected to total body 6Gy ɣ-irradiation. After 6 h, mice were sacrificed, thymocytes isolated and analyzed by FACS. Mean ± SD is shown (Two-tailed unpaired *t*-test, (**E**) *n* = 20, DF = 18, t = 2.91 *p* = 0.0093; (**F**) *n* = 13, DF = 11, t = 0.4482, n.s.). (**G**,**H**) Fibrosarcoma cisplatin sensitivity in Mdm4^Tg15^ (**G**) and WT females (**H**) ± ICI. After reaching a volume of 200 mm^3^, animals were treated with bi-weekly i.p. cisplatin (5 mg/Kg) and weekly s.c. ICI or vehicle (PBS). Tumor size is relative to the volume at the first treatment (two-way ANOVA not RM, (**G**) DF = 1, F_(treatment)_ = 7029; (**H**) DF = 1, F_(treatment)_ = 0003, n.s.).

**Figure 3 cancers-11-01349-f003:**
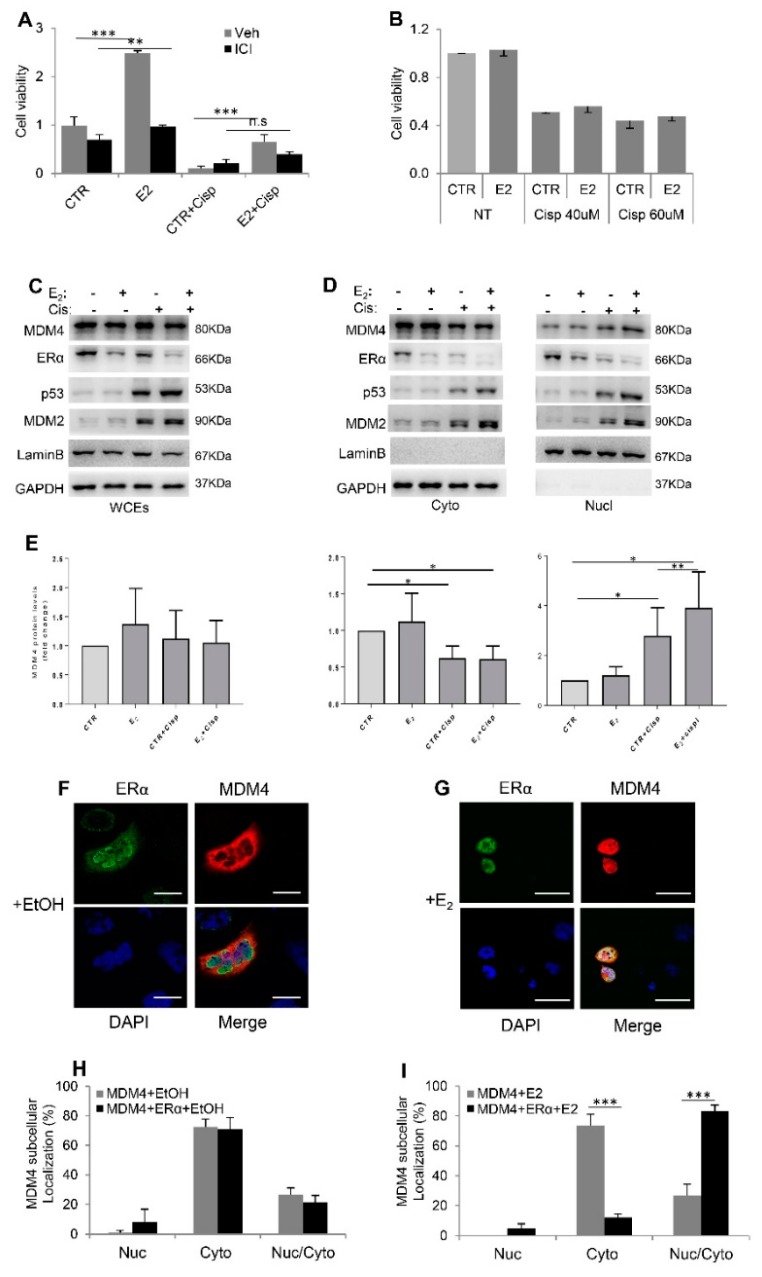
E_2_ alters MDM4 subcellular localization. (**A**,**B**) Relative cell viability of MCF-7 (**A**) and MDA-MB-231 (**B**) by cell titer assay. Graphs show mean ± SD of two experiments performed in tetraplicate (Two-tailed unpaired *t*-test, (**A**) CTR vs. E_2,_ t = 13.45, df = 4, *** *p* = 0.0002, CTR + Cisp vs. E_2_ + Cisp, t = 18.29, df = 8, *** *p* < 0.0001, CTR + ICI vs. E_2_ + ICI, t = 4.834 df = 6, ** *p* = 0.0029, CTR + Cisp + ICI vs. E_2_ + Cisp + ICI, t = 2.284, df = 6, *p* = 0.0624). (**C**,**D**) WB analysis of whole cell extracts (WCEs) (**C**) and cytoplasmic (Cyto) and nuclear extracts (Nucl) (**D**) of MCF-7 cells treated as indicated. Laminin and GAPDH were used as LC. (**E**) MDM4 levels under indicated treatments, relative to the untreated cells (CTR) arbitrarily set to 1 and corrected to the respective loading control. Mean ± SD of 4 independent biological replicates is shown (One sample *t*-test, Cyto graph: CTR + Cisp vs. CTR, DF = 3, t = 4.48, * *p* = 0.021, E_2_ + Cisp vs. CTR, DF = 3, t = 4,328, * *p* = 0.023; Nucl graph: CTR + Cisp vs. CTR, DF = 3, t = 3198, * *p* = 0,0494, E_2_ + Cispl vs. CTR, DF = 3, t = 4.034, * *p* = 0.0274; E2 + Cisp vs. CTR + Cisp Paired *t*-test DF = 3, t = 6.451, ** *p* = 0.0076). (**F**,**G**) Representative pictures of ERα (green) and MDM4 (red) exogenously expressed in A2780 cells treated as indicated for 20 h. Nuclei were counterstained with DAPI (blue) (scale bars, 50 µm). (**H**,**I**) Graphs show the percentage of A2780 cells showing only Nuclear (Nuc) or Cytoplasmic (Cyto) or Nuclear and Cytoplasmic (Nuc/Cyto) MDM4 signal relative to the total number of positive cells. Mean ± SD of 3 independent biological replicates is shown (Two-tailed unpaired *t*-test, (**I**) Cyto MDM4 + E_2_ vs. Cyto MDM4 + ERα + E_2,_ t = 13.13, df = 4, *** *p* = 0.0002, Nuc/Cyto MDM4 + E_2_ vs. Nuc/Cito MDM4 + ERα + E_2,_ t = 11.08, df = 4, *** *p* = 0.0004).

**Figure 4 cancers-11-01349-f004:**
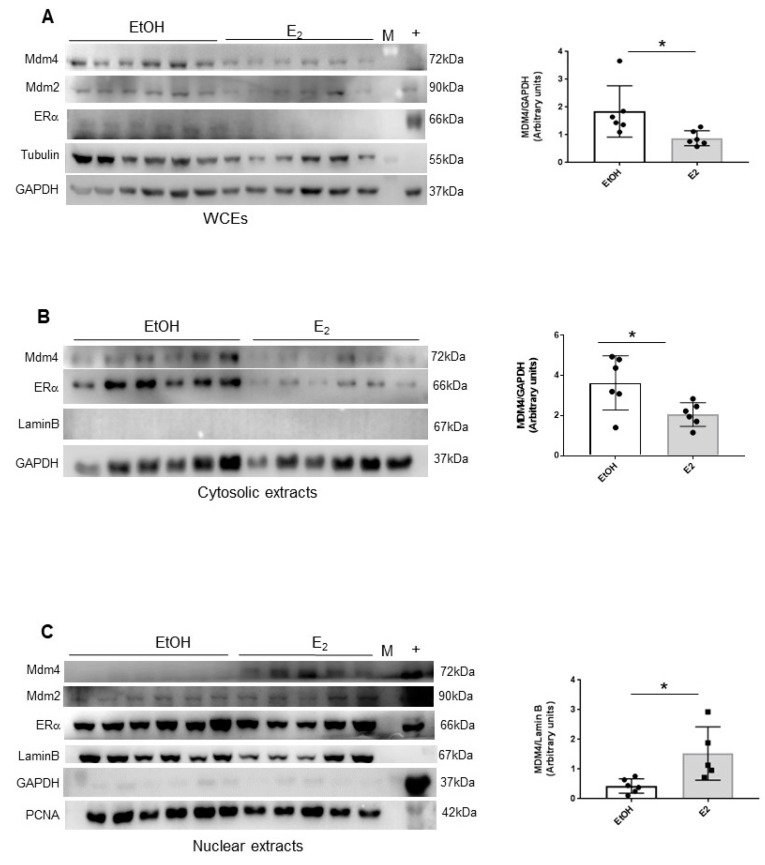
E_2_ alters the subcellular localization of Mdm4 in mouse thymocytes. (**A**–**C**), WB analysis of whole cell extracts (WCEs) (**A**) cytosolic extracts (**B**) and nuclear extracts (**C**) of thymocytes of Mdm4^Tg15^ females treated with a single i.p. dose of physiologic solution or 50 µg/Kg E_2_ and sacrificed after 16 h. Laminin and GAPDH were used as LC. Graphs show mean ± SD of MDM4 levels quantified by Alliance V_1607 software (two-tailed unpaired *t*-test, WCEs: t = 2.443, df = 10, * *p* = 0.0346, Cytosolic extracts: t = 2.629, df = 10, * *p* = 0.0252, Nuclear extracts: t = 2.891, df = 9, * *p* = 0,0179).

**Figure 5 cancers-11-01349-f005:**
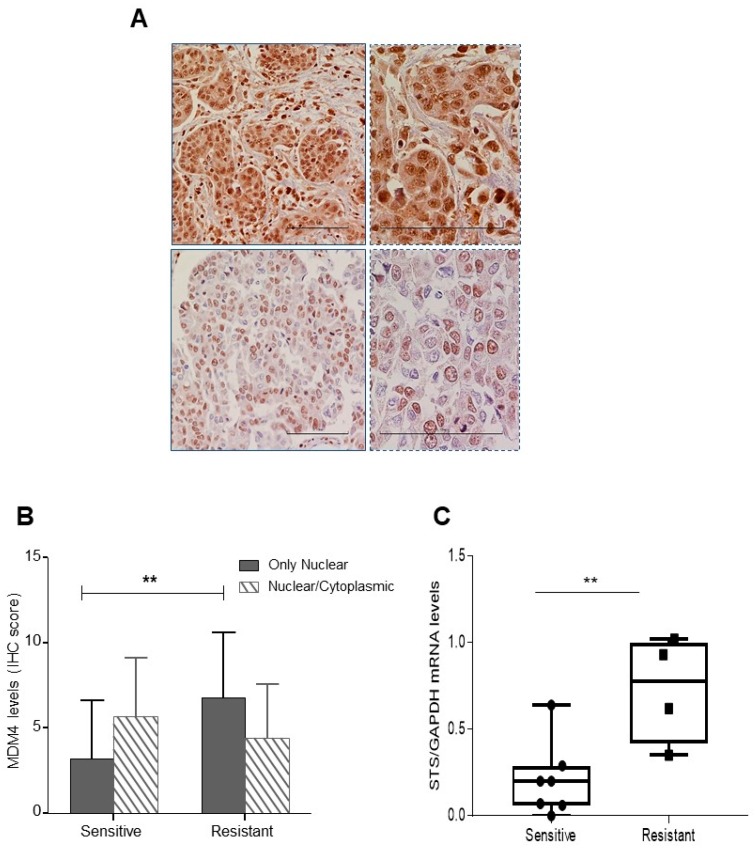
Nuclear MDM4 signal is increased in chemo-resistant high-grade serous ovarian cancer (HGSOC). (**A**) Representative pictures of MDM4 showing nuclear/cytoplasmic (upper panel) and only nuclear (lower panel) signal in HGSOC, irrespective of their IRS score. Right panels represent magnification of left panels (Scale bar, 100 µm). (**B**) MDM4 levels (mean ± SD, IHC = immunoreactive score) in nuclear and nuclear/cytoplasmic compartments of sensitive and resistant patients (** *p* < 0.01, two-tailed unpaired *t*-test, *n* = 33, DF = 31, t = 2.773, ** *p* = 0.0093). (**C**), qPCR of STS mRNA extracted from available matched frozen samples examined in B (mean ± SD, two-tailed unpaired *t*-test, *n* = 11, DF = 9, t = 3.34, ** *p* = 0.0087).

**Figure 6 cancers-11-01349-f006:**
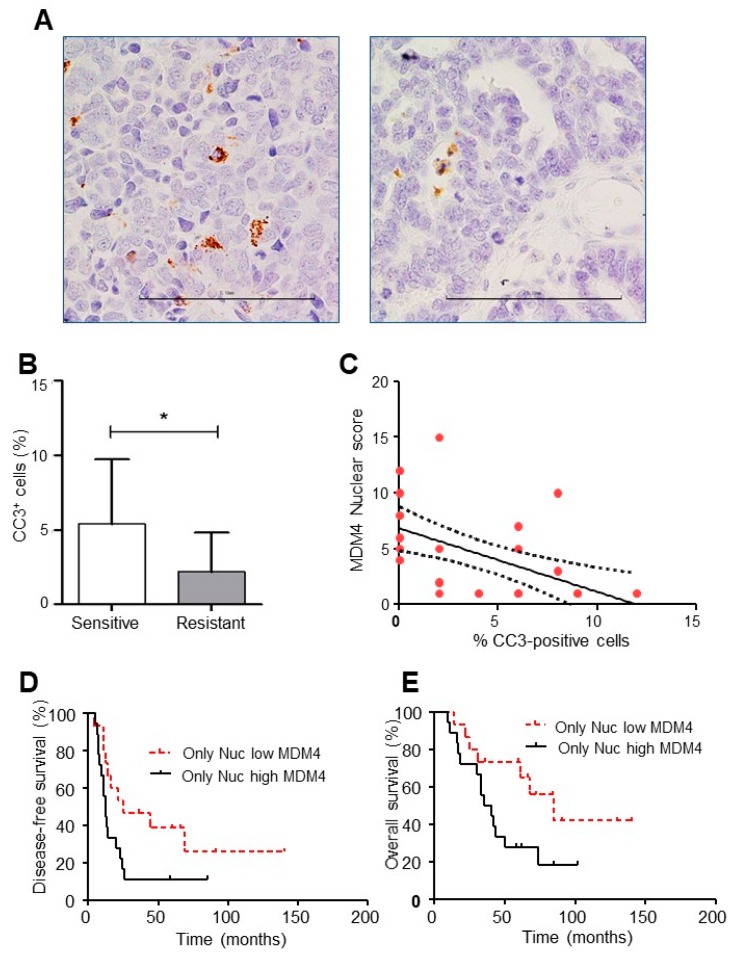
Nuclear MDM4 correlates with decreased cell death and DFS in HGSOC. (**A**) Representative pictures of cleaved caspase-3 staining (CC3) in HGSOC (left and right panels show high and low expression levels, respectively, scale bar 100 µm). (**B**) CC3 expression in sensitive and resistant patients (mean ± SD, two-tailed unpaired *t*-test, *n* = 31, DF = 29, t = 2.230, * *p* = 0.034). (**C**), Correlation between CC3 and nuclear MDM4 in HGSOC (Spearman rank correlation, *p* < 0.0001, r = 0.7, 95% CI −0.8 to 0.4). (**D**,**E**) Disease-free survival (**D**) and overall survival (**E**) of patients according to the expression of nuclear MDM4 in HGSOC. High expression of MDM4 in the nuclear compartment is associated with decreased disease-free survival and overall survival (Log-rank test (**D**) *χ^2^*= 4.172, df = 1 *p* = 0.041; (**E**) *χ^2^*= 3.670, df = 1 *p* = 0.055).

**Table 1 cancers-11-01349-t001:** P53 status in fibro-sarcoma of Mdm4^Tg^ and WT mice.

	Mdm4^Tg^		WT	
P53 Status	♀	♂	♀	♂
WT	3 ^#^	2	3	2
Mut	4	3	4	1

^#^ Number of animals.

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
