# Peer review of "Estrogens Counteract Platinum-Chemosensitivity by Modifying the Subcellular Localization of MDM4"

_cancers, 2019, doi:10.3390/cancers11091349_

Round 1
Reviewer 1 Report
This manuscript addresses the dichotomous activities of MDM4: it promotes cell proliferation and accelerates formation of fibrosarcoms; on the other hand it enhances sensitivity to DNA damage caused by agents such as cisplatin and gamma radiation. Using MDM4 transgenic animal models, the authors show that Mdm4 enhances chemo and radio sensitivity in males, but not in female animals, where as the tumor p-promoting activity is not gender-dependent. Furthermore abrogation of estrogen signaling (using ER alpha degrades fuvestrant) in female mice made them sensitive to cisplatin treatment. These data are important in understanding mechanisms of MDM4 function and their therapeutic implications in cancer.
Overall, the experimental data are of high quality. The manuscript is well written. However, there are some important concerns that need to addressed:
1. The immunofluorescence shown in Figure 3F is not of sufficient quality and rigor. Panel with DAPI staining alone should be shown. Immunofluorescence studies with MDM4 antibodies should include a control panel of cells where MDM4 is depleted with si/shRNA against MDM4.
2. The authors should look into the published literature and cite publications where MDM4 and ER alpha relationship (including interaction between the two proteins) was reported. The results of the current study should be discussed and interpreted taking into consideration what is already known.
3. The description of methodology for immunohistochemical analysis of ovarian cancer patient tumors should include more specifics as to if the analysis was done on sinle sample fro each tumor or biological duplicates or triplicates. Was the immunohistochemistry performed on individual tumors or where the tumor tissues were on tissue microarrays (TMAs)?
Minor comments:
On page 5, sentence 167: ‘estrogen inhibitor fulvestrant’ should be corrected to ‘estrogen receptor inhibitor fulvestrant’.
Page 11, sentence 325: CCL3 should be changed to CC3.
Reviewer 2 Report
The authors report that MDM4 overexpression in mice can promote 3MCA-induced sarcoma formation, but sensitizes the tumors to cisplatin. They further found that the latter effect appears to be limited in males, and treating tumors from female MDM4 mice with an estrogen inhibitor re-sensitizes the tumors to cisplatin. Mechanistically, they show that estrogen can drive nuclear localization of MDM4. They also show the clinical significance of their finding by looking at correlation between MDM4 nuclear localization and platinum sensitivity in a small cohort of ovarian cancer patients. Overall, the study is well designed, and provides a new insight into the oncogenic/tumor-suppressing function of MDM4. The following concerns/questions need to be addressed:
Table 1, it is not clear what those numbers indicate. This table is very confusing, and needs to be modified.
It is interesting that the tumors from MDM4 wt and Tg mice have a similar rate of p53 inactivation. While this suggests that MDM4 expression did not reduce the selective pressure to inactivate p53 as indicated by the authors, it does not seem to be right to conclude the MDM4-mediated effect is independent of p53.
Fig 2, as cisplatin can induce p53, the responses of sarcomas to cisplatin might be influenced by the p53 mutation status of the tumors. Therefore, the authors need to indicate whether there is a difference in the p53 mutation rate between MDM4wt and Tg tumors in both male and female mice, and probably separate p53-wt and mutated tumors for their responses to cisplatin.
Fig 2B, the tumor growth still has some difference between wt and Tg tumors in female mice, although the difference might be insignificant. Fig 2G/H, it would be necessary to test ICI in male wt/Tg mice to rule out the possibility that the observed effect is due to ER-independent activity. On the other hand, while the difference shown in Fig 2G is marginal, MDM4 Tg tumors appears to be resistant to cisplatin if combining Fig 2G and 2H - this is different from what was shown in Fig 2B. Can the authors explain the discrepancy?
Fig 3D, estrogen is supposed to bind ER and increase ER nuclear translocation. This is not shown in the panel.
Fig 3F/G show very few cells exclusively express MDM4 in their nuclei. However, MDM4 exclusive-nuclear staining appear to occur in a large number of ovarian cancer samples shown in Fig 5A and 5B. Can the authors give an explanation?
Fig 5A shows that those cells with exclusively nuclear staining often have much weak MDM4 staining (lower) as compared to the samples claimed to be nuclear/cytoplasmic staining (upper). There is a possibility that the short time of DAB exposure (which yield weak staining) results in no staining in the cytoplasm. Please clarify this technical issue.
Fig 6C, what is the correlation coefficient (R)? Only show the p value is not sufficient.
Round 2
Reviewer 1 Report
Unfortunately, the authors have not appropriately addressed the concern #1. That the authors used an antibody (purchased from Origene) recognizes only exogenously overexpressed MDM4, but not endogenous MDM4 protein, is not a justifiable scientific response to the query on the issue of lack of sufficient quality and rigor for the immunofluorescence data shown in Figure 3 F & G. MDM4 antibodies that recognize endogenous MDM4 in immunoflourescence (IF) study are commercially available. It is unclear (rather puzzling) why the authors chose exogenous overexpression of MDM4 to analyze the effect of estrogen on subcellular localization of ER alpha and MDM4 rather than analyzing the effect of estrogen on subcellular localization of the endogenous ER alpha and MDM4 proteins. Overexpression of proteins complicates the interpretation of subcellular localization data. Data for figures 3 A, B, C, D and E were obtained using MCF 7 cells. Therefore it would be logical to show IF data using these cells rather than switching to A2780 ovarian cancer cells. The authors should show IF data on the effect of estrogen on subcellular distribution endogenous ER alpha and MDM4 proteins (using an antibody that recognizes endogenous MDM4) in MCF7 cells that has sufficient levels of endogenous ER alpha and MDM4 proteins (based on Figures 3C & D).
The statement in Lines 236 and 237: ‘ERα localization was not substantially altered by MDM4 expression but its increased nuclear localization following E2 treatment (Figure S4A and B)’ does not agree with the data shown in Supplementary Figure S4 A & B. Looking at these figures, there is no clear difference in nuclear localization of ER alpha as well as MDM4 with or without treatment with estrogen.
The authors have addressed concerns 2 and 3.
Reviewer 2 Report
The authors have addressed my previous comments.
Author Response
We thank the reviewer.
Round 3
Reviewer 1 Report
The authors should incorporate into "discussion" section their explanation for MCF7 cells not being a good system to analyze MDM4 localization quantitatively by immunofluorescence: "because in MCF7 a fraction of MDM4 is in the nucleus (Mancini et al., EMBO J 2009; Figure 3C), a feature that frequently occurs in human tumor cells (Gembarska et al., Nat Med 2012). This condition impairs the ability of Immunofluorescence to efficiently detect quantitative changes of MDM4 signal in the nucleus upon estradiol treatment".
Author Response
According to the Reviewer's suggestion, we have included the indicated sentence in the text (Page 7, lines 240-242).